# SMAD4 and the TGFβ Pathway in Patients with Pancreatic Ductal Adenocarcinoma

**DOI:** 10.3390/ijms21103534

**Published:** 2020-05-16

**Authors:** Julie Dardare, Andréa Witz, Jean-Louis Merlin, Pauline Gilson, Alexandre Harlé

**Affiliations:** Université de Lorraine, CNRS UMR7039 CRAN, Service de Biopathologie, Institut de Cancérologie de Lorraine, 54519 Vandoeuvre-lès-Nancy, France; j.dardare@nancy.unicancer.fr (J.D.); a.witz@nancy.unicancer.fr (A.W.); jl.merlin@nancy.unicancer.fr (J.-L.M.); p.gilson@nancy.unicancer.fr (P.G.)

**Keywords:** SMAD4, TGFβ, pancreatic cancer, PDAC, EMT

## Abstract

Pancreatic ductal adenocarcinoma (PDAC) is the fourth leading cause of cancer death worldwide. PDAC is an aggressive disease with an 11-month median overall survival and a five-year survival of less than 5%. Incidence of PDAC is constantly increasing and is predicted to become the second leading cause of cancer in Western countries within a decade. Despite research and therapeutic development, current knowledge about PDAC molecular mechanisms still needs improvements and it seems crucial to identify novel therapeutic targets. Genomic analyses of PDAC revealed that transforming growth factor β (TGFβ) signaling pathways are modified and the *SMAD4* gene is altered in 47% and 60% of cases, respectively, highlighting their major roles in PDAC development. TGFβ can play a dual role in malignancy depending on the context, sometimes as an inhibitor and sometimes as an inducer of tumor progression. TGFβ signaling was identified as a potent inducer of epithelial-to-mesenchymal transition (EMT), a process that confers migratory and invasive properties to epithelial cells during cancer. Therefore, aberrant TGFβ signaling and EMT are linked to promoting PDAC aggressiveness. TGFβ and SMAD pathways were extensively studied but the mechanisms leading to cancer promotion and development still remain unclear. This review aims to describe the complex role of SMAD4 in the TGFβ pathway in patients with PDAC.

## 1. Introduction

Pancreatic ductal adenocarcinoma (PDAC) is the most common malignancy of the pancreas, with an extremely poor prognosis with a five-year survival rate of 5% and a median survival of less than 11 months [1]. Due to the lack of specific clinical symptoms and biomarkers for PDAC detection, as well as the difficulty in imaging the tumors in early stages, PDAC is often diagnosed in the later stages [2]. PDAC is currently the fourth leading cause of cancer death worldwide and is projected to become the second leading cause of cancer-related deaths before 2030 in the United States [3]. 

PDAC is an aggressive disease with early distant metastasis, as well as perineural and vascular local growth, making it eligible for surgical resection in only 10%–20% of patients [4]. Moreover, 69% to 75% of patients that benefitted from surgery finally relapse within two years, and 80% to 90% relapse within five years [4]. For patients with unresectable or potentially resectable PDAC, most common therapies used in cancer treatment including chemotherapy or radiotherapy are limited due to a remarkable treatment resistance. Gemcitabine was the standard of care for more than two decades in patients with pancreatic cancer, after demonstrating significant improvement in overall survival (5.6 months as compared with 4.4 months for fluorouracil-based regimen; *p* = 0.0022) [5]. The FOLFIRINOX protocol including fluorouracil, leucovorin, irinotecan, and oxaliplatin is now the first-line option for patients with metastatic pancreas cancer or in adjuvant settings with a better efficacy compared to gemcitabine [6,7]. Nab-paclitaxel also represents an alternative in association with gemcitabine in metastatic PDAC patients, with a median overall survival of 8.5 months (hazard ratio (HR) 0.72; 95% confidence interval (CI) 0.62, 0.83; *p* < 0.001) [8]. 

The POLO clinical trial (Pancreas Cancer Olaparib Ongoing) evaluated the efficacy of maintenance therapy with olaparib, a poly(adenosine diphosphate–ribose) polymerase (PARP) inhibitor, in patients with germline *BRCA* mutation and metastatic pancreatic cancer. Median progression-free survival was significantly better in the olaparib group compared to placebo (7.4 months versus 3.8; HR 0.53; 95% CI 0.35, 0.82; *p* = 0.004) [9]. However, targeted therapies still have a limited role due to the lack of understanding of the complex molecular biology of PDAC.

Genomic analyses of PDAC revealed a large panel of molecular alterations particularly affecting Kirsten rat sarcoma viral oncogene (*KRAS*), tumor protein P53 *(TP53)*, Mothers Against Decapentaplegic Homolog 4 (*SMAD4)*, and cyclin-dependent kinase inhibitor 2A *(CDKN2A)* genes [10,11]. *SMAD4* gene is inactivated in approximately 60% of PDAC cases [12], and it is an effector of the transforming growth factor β (TGFβ) signaling pathway which is also altered in 47% of PDAC cases [13]. Given the multiple roles of TGFβ in cancer and its impact on PDAC, it seems interesting to focus on its effector SMAD4. This review aims to summarize current knowledge concerning SMAD4 in TGFβ pathway in patients with PDAC and to discuss the potential SMAD4 molecular targeting in therapeutic. 

## 2. Methods

### Search Strategy 

This review was conducted through a systematic review according to the directions denoted by the Preferred Reporting Items for Systematic reviews and Meta-Analysis (PRISMA). To investigate the entirety of the published studies on SMAD4 and the TGFβ pathway in patients with pancreatic ductal adenocarcinoma, a comprehensive literature search of the electronic database PubMed was performed up to April 2020. Studies were selected using the following search terms: “PDAC” and “TGFβ” and “SMAD4”. 

## 3. Results

### 3.1. TGFβ Signaling Pathways

TGFβ is a ubiquitously expressed cytokine belonging to a family composed of two branches: the TGFβ branch, represented by ligands such as TGFβ, activin, nodal, or myostatin, and the bone morphogenetic protein (BMP) branch, represented by ligands such as BMP, and growth and differentiation factor (GDF) [14]. TGFβ firstly provides a versatile means of driving developmental programs in mammalian; then, it acts in adult homeostasis with regulation of tissue repair, wound healing, and immune response. There is also a wide panel of cell-type specific biological TGFβ activities such as differentiation, cell-cycle arrest, migration, adhesion, apoptosis, or cancer biology [15,16,17]. TGFβ is also a potent inducer of epithelial-to-mesenchymal transition (EMT), a well-coordinated process during embryonic development and a pathological feature in neoplasia and fibrosis [15]. TGFβ signaling pathway is activated through either a SMAD-dependent or a SMAD-independent process. 

#### 3.1.1. The Canonical TGFβ Signaling Pathway

TGFβ signaling is mediated in most cells through three cell-surface proteins: the two serine/threonine kinase receptors TGFβ receptor I (TβR-I) and TGFβ receptor II (TβR-II), and the TGFβ receptor III (TβR-III). TGFβ ligands can bind directly to TβR-II. TβR-II is constitutively active and can phosphorylate TβR-I, leading to its activation and the propagation of the signal through the phosphorylation of SMAD proteins in their C-terminal serines (SXS) motif [16]. The protein phosphatase 2a (PP2a) can be recruited to dephosphorylate TβR-I, thereby preventing further signaling [18]. TβR-I is able to activate different members of the receptor regulated SMAD (R-SMAD) family [19]. In the presence of ligands from the TGFβ branch, TβR-I induces the activation of SMAD2 and 3, whereas it activates SMAD1, 5, and 8 in the presence of ligands from the BMP branch [20]. Recognition of R-SMADs may be facilitated by auxiliary proteins, such as the SMAD anchor for receptor activation (SARA) that immobilizes SMAD2 and SMAD3 near the cell surface [21]. Once activated, the R-SMADs undergo a change of conformation that increases their affinity for SMAD4, a co-mediator SMAD (Co-SMAD) distributed throughout the cytoplasm. R-SMAD and Co-SMAD form a heterotrimeric complex, which is translocated and accumulated into the nucleus. SMAD proteins act as transcription factors by binding to the SMAD binding element (SBE) due to their Mad-Homology 1 (MH1) domain [21]. SMAD activates transcriptional TGFβ responses through the Mad-Homology 2 (MH2) domain [22] in association with a transcriptional co-activator, such as CREB Binding Protein (CBP)/p300 (Figure 1) [20,21,23]. 

Activated R-SMADs can be submitted to a second phosphorylation in their interdomain linker region by cyclin-dependent kinases (CDK) 8 and 9, which favors interactions with a co-activator and enhances their transcriptional activity [19]. On the other hand, CDK8 and CDK9 negatively regulate SMAD levels by recruiting glycogen synthase kinase 3 (GSK3) for proteasomal degradation by E3 ubiquitin ligases [24]. This SMAD-dependent signaling pathway can also be negatively regulated by the inhibitory SMAD (I-SMAD) family, including SMAD6 and 7. I-SMADs are expressed through ligand-induced signaling and act as an auto-inhibition feedback to regulate TGFβ signaling pathway [14]. SMAD7 expression can also be regulated through activation of epidermal growth factor receptor (EGFR), interferon γ signaling by STAT (signal transducer and activator of transcription) protein, or activation of NF-κB by Tumor Necrosis Factor α (TNFα) [14,25,26]. SMAD7 binds to the activated receptors in competition with R-SMADs and leads to ubiquitination and degradation of the receptors with the help of E3 ubiquitin ligases termed SMAD ubiquitination regulatory factors (Smurfs) 1 and 2 [21]. Smurfs also interact directly with R-SMADs in the cytoplasm; Smurf 1 is more specific to BMP responses by targeting SMAD1 and SMAD5, whereas Smurf 2 has the ability to interact with all R-SMADs [25]. SMAD6 can also negatively regulate TGFβ signaling pathway through interfering with the heterotrimerization of the BMP-activated SMADs and SMAD4 [25]. 

#### 3.1.2. Non-canonical TGFβ Signaling Pathway

SMAD signaling is crucial for most but not all TGFβ-regulated transcriptional responses. SMAD-independent signaling cascades can also operate in a context-dependent manner including the mitogen-activated protein kinase (MAPK) and the phosphatidylinositol 3-kinase (PI3K) pathways, as well as small GTPases [25,27,28]. 

The activation of TβR-I phosphorylates Src homology domain 2-containing protein (Shc) and induces its association with the adaptor protein growth factor receptor-binding protein 2 (Grb2) and the GTP exchange factor SOS. This complex enables Ras to become its active form that can enhance the kinase cascade with c-Raf, MEK1 or MEK2, and Erk1 or Erk2 [29]. TGFβ also induces p38 and JNK pathways through the activation of the MAP kinase kinase kinase (MAP3K) TGFβ-activated kinase 1 (TAK1), which activates the MAP kinase kinases (MKKs) MKK3/6 and MKK4, respectively. TGFβ receptors induce the ubiquitination of the TNF receptor-associated factor 6 (TRAF6), which is a critical activator of TAK1. Then, TAK1 can activate p38 and JNK MAPK pathways [30,31]. TGFβ signaling pathway may induce NF-κB signaling due to TAK1 phosphorylation and activation of inhibitor of nuclear factor kappa B (IκB) [25]. TβR-II was shown to be a target for Src, which is an oncoprotein able to phosphorylate TβR-II on a tyrosine residue. Src coordinates the docking of Grb2 and Shc, and it contributes in this manner to TGFβ-induced activation of the p38 MAPK pathway [30,32]. 

In response to TGFβ, TβR-I was shown to interact with the catalytic subunit of PI3K, leading to the activation of Akt and mTOR (mammalian target of rapamycin) downstream proteins [18,27]. mTOR serine/threonine protein kinase can phosphorylate the two principle effectors S6 kinase 1 (S6K1) and the eukaryotic initiation factor 4E-binding protein 1 (4E-BP1), both acting on the translation of proteins that contribute to EMT [33].

TGFβ can activate Rho-like GTPases including RhoA, Rac1, and Cdc42 [25]. Rac and Cdc42 interact with MAP3K, which regulates JNK and p38 signaling [25]. Rho GTPases are required for the organization of the microtubules and actin cytoskeletons. Therefore, the TGFβ-induced activation of the small GTPases seems to contribute to the regulation of cell adhesion and migration [18]. Indeed, Rho and Rac were identified to mediate the pro-EMT, pro migratory, and invasive effects of TGFβ in normal and in cancer epithelial cells [34]. 

It appears clear that the non-canonical TGFβ pathway is linked to cancer features such as migration, invasion, and EMT. Interestingly, it was described that the non-canonical TGFβ pathway is inappropriately activated in metastatic cancer cells and interferes with the SMAD2/3-mediated tumor suppressing message [17].

### 3.2. TGFβ and PDAC

TGFβ was identified as a major signaling pathway in PDAC. Recently, a study indicated that elevated serum soluble TGFβ levels at diagnosis predict poor progression-free survival (PFS) and overall survival (OS) in unresectable pancreatic cancer compared to patients with a low level of soluble TGFβ (OS 13.7 vs. 9.2 months; HR 2.602; *p* = 0.004; PFS 9.0 vs. 5.8 months; HR 2.010; *p* = 0.034) [35]. Another study in patients with locally advanced and metastatic pancreatic cancers reported that higher soluble TGFβ level exhibited poor OS than lower serum soluble TGFβ (HR 1.35; 95% CI 1.07, 1.69; *p* = 0.011) [36]. These observations corroborate the “TGFβ paradox”; TGFβ can be both a powerful tumor suppressor and a tumor promoter in a context-dependent manner [15].

#### 3.2.1. TGFβ as a Tumor Suppressor

In normal and premalignant cells, TGFβ plays a role in homeostasis and promotes tumor suppression directly through cellular autonomous effects such as cytostasis, differentiation, or apoptosis (Figure 2) [15]. When cell proliferation rate is high, the TGFβ pathway is involved in cytostatic effects, stimulates cellular differentiation, or triggers proapoptotic mechanisms. In cell-cycle regulation, TGFβ suppresses the expression of CDKs that are crucial for Gap 1 (G1) phase progression, TGFβ also enhances the expression of CDK inhibitors (CDKIs) such as p15 and p21. TGFβ represses the expression and function of c-Myc, a well-known transcription factor implied in cell proliferation [15,16]. TGFβ also drives precursor cells into a less proliferative stage, or it can lead cells to apoptosis [37]. 

It was suggested that the tumor-suppressive role of TGFβ is only effective when the TGFβ signaling pathway is not defective [38]. However, in PDAC alterations of TGFβ signaling through the mutation of genes involved in the pathway (e.g., *SMAD4, SMAD3, TβR-I, TβR-II, ACVR1B*, and *ACVR2A*), this role is present in 47% of cases [13], highlighting that TGFβ can sometimes acts as a tumor promoter.

#### 3.2.2. TGFβ as a Tumor Promoter

TGFβ signaling was identified as a tumor suppressor in normal pancreatic cells and in stages I and II PDAC by inhibiting cell proliferation, but it also showed tumorigenic activity in late-stage malignancy [39]. A high level of TGFβ signaling in PDAC was associated with poor prognosis despite its clear tumor-suppressive effect [40].

In non-cancer situations, TGFβ initially prevents premalignant cells through its tumor suppressor activities; however, in advanced stages of malignant cancer, cells can turn TGFβ signaling to their advantage [15]. TGFβ indirectly promotes tumor suppression by acting on the cellular micro environment; however, in PDAC, TGFβ was shown to induce stromal proliferation in the tumor microenvironment and promote EMT (Figure 2) [41,42,43]. 

Pancreatic cancer is characterized by a desmoplastic stroma enriched with small blood vessels and extracellular matrix proteins partly produced by cancer-associated fibroblasts (CAFs) and inflammatory cells [44]. TGFβ contributes to PDAC desmoplasia by enhancing the conversion of fibroblasts or endothelial cells into myofibroblasts also known as CAFs [45,46]. Aggressiveness is amplified by immune cells and fibroblasts in the tumor microenvironment, which can produce high levels of TGFβ in a paracrine manner [46,47]. TGFβ can induce proangiogenic factors such as vascular endothelial growth factor (VEGF), allowing PDAC progression, invasion, and metastases [46].

PDAC-infiltrating lymphocytes were shown to improve prognosis for patients with high rates of cluster of differentiation 4 (CD4), CD8, and dendritic cells [48]. As an immunosuppressive cytokine, TGFβ inhibits the function and development of innate and adaptive immune systems including macrophages, natural killers, dendritic cells, and T cells. TGFβ also stimulates regulatory T cells, which inhibit the function of other lymphocytes [15,47,49]. These suppressive functions confer to TGFβ one of many cancer hallmarks with avoiding immune destruction. On the other hand, others suggested that defective TGFβ signaling, as observed in PDAC, could contribute to an excessive inflammation in favor of tumor progression [15,47].

TGFβ triggers EMT through induction of the expression of specific transcription factors Snail and Zeb1/2 [50]. EMT provides migratory and invasive behaviors to the cells due to cell adhesion modifications. This process involves a loss of epithelial features (i.e., E-cadherin and other components of epithelial cell junctions) and the acquisition of mesenchymal features leading to motility and invasive properties [50]. In PDAC, EMT represents an important process leading to the progression and metastasis of cancer cells. 

### 3.3. SMAD4

SMAD4 or DPC4 (deleted in pancreatic cancer locus 4) is an intracellular transcriptional mediator of the TGFβ signaling pathway encoded by the *SMAD4* gene which is located on the 18q21.2 human chromosome locus and is composed of 12 coding exons and 10 introns (GRCh38, NM_005359.5). The SMAD4 protein has a molecular weight of 60 kDa and is composed of 552 amino acids [12,51]. SMAD4 is composed of MH1 and MH2 domains connected by a linker region involved in DNA binding and its transcriptional activity [51]. 

SMAD4 activity and stability are regulated by three major signaling pathways through its linker region that contains a growth factor-regulated transcription activation domain integrating Wnt, fibroblast growth factor (FGF)/epidermal growth factor (EGF), and TGFβ signaling [52]. SMAD4 can be phosphorylated and activated through the Erk pathway in response to FGF and EGF. However, this MAPK pathway triggers phosphorylation of SMAD4 by GSK3, which creates a phosphodegron leading SMAD4 to its polyubiquitination and degradation by the E3-ligase β-TrCP. The Wnt pathway can regulate the phosphodegron, and it inhibits the binding of β-TrCP to SMAD4. In the presence of Wnt, SMAD4 phosphorylation by GSK3 is inhibited and TGFβ signaling is consequently prolonged [52].

### 3.4. SMAD4 and PDAC

The most common driver mutational events in PDAC occur in four cancer genes *KRAS*, *TP53*, *SMAD4*, and *CDKN2A*. *KRAS* is activated by mutation in almost 90% of cases, *TP53* is inactivated in 50%–75% of cases, and *CDK2NA* is inactivated in about 30% of patients with PDAC [10,53]. *SMAD4* was firstly identified as a tumor suppressor gene that is inactivated in 60% of PDAC cases due to homozygous deletion or mutation [12]. The *SMAD4* gene is located on the q arm of chromosome 18, which is frequently deleted hemi- or homozygously in PDAC [54]. Alterations of the *SMAD4* gene occur in late stages when the carcinoma is histologically recognizable [55]. In pancreatic cancers, homozygous deletion of SMAD4 is found in approximately 30% of cases, inactivation is found in 20% of cases, and allelic loss of chromosome 18q is found in almost 90% of cases [56]. Hotspot mutations (i.e., nonsense, missense, or frameshift indels) in the C-terminal MH2 domain of *SMAD4* were described [56]. Nonsense substitutions were identified in 25% of cases, missense substitutions were identified in 37% of cases, and frameshift insertion and deletion were identified in 7% and 5% of cases, respectively (GRCh38. COSMIC v91). *SMAD4* mutations lead to a non-functional or truncated SMAD4 protein or to the protein degradation through the ubiquitin–proteasome pathway [57]. Notably, two SMAD4 missense mutations (Pro130Ser and Asn351His) were found to increase SMAD4 phosphorylation by GSK3 and lead to protein degradation [57]. 

#### 3.4.1. SMAD4 and EMT 

Loss of SMAD4 inhibits the tumor suppressor effects of TGFβ without affecting tumor response, promoting a more aggressive phenotype [58]. *SMAD4* knockdown in PDAC cell lines showed SMAD4 as a regulator of genes involved in cell proliferation, adhesion, and motility [59]. In another study using *SMAD4* silencing, SMAD4 was required in TGFβ-induced cell-cycle arrest and migration. Regarding EMT, authors found that *SMAD4* silencing did not abolish EMT, suggesting that EMT can be conducted not only through SMAD4 signaling [58].

TGFβ can induce EMT in a SMAD4-dependent manner through the expression of Snail and Zeb1 transcription factors and the decrease of E-cadherin epithelial marker [60]. SMAD4 deficiency in PDAC cells induces E-cadherin expression and is associated with an ineffectiveness of TGFβ to induce EMT [61]. However, a study of patients with PDAC examined the clinical association between SMAD4 expression and EMT status and revealed that SMAD4 suppression was associated with a mesenchymal phenotype [62], highlighting the complex mechanistic of EMT. An increase in in the non-canonical TGFβ pathways was demonstrated in SMAD4-defective PDAC cells [63]. This SMAD4-independent pathway activates the PI3K/Akt/mTOR pathway and induces small GTPases, which can contribute to EMT [33,34,53]. SMAD4 can suppress JNK activation; therefore, in the case of SMAD4 deletion, JNK activity is subsequently enhanced [64] and can be required in the EMT response [65]. 

TGFβ-induced EMT is commonly linked with tumor progression; however, a study demonstrated that TGFβ can also induce a SMAD4-dependent lethal EMT in PDAC cells. In SMAD4 wild-type cells, TGFβ induces the conversion of the transcription factor Sox4 from an enhancer of tumorigenesis into a promoter of apoptosis. In this process, Snail induces the repression of Klf5, which is associated with Sox4 to prevent Sox4-mediated apoptosis in oncogenesis [42]. This SMAD4-dependent mechanism can provide an explanation about the dual role of TGFβ in tumors.

#### 3.4.2. Effect of SMAD4 Expression in Patients with PDAC

Loss of SMAD4 expression occurs late in PDAC tumor progression [62] and is correlated with a higher metastatic burden [66], as well as malignant phenotypes of PDAC, including degree of differentiation, tumor size, and lymph node metastasis [67]. Patients with SMAD4 loss are commonly associated with distant disease progression, whereas patients with wild-type SMAD4 harbored a local tumor pattern of progression [62].

The prognostic effects of SMAD4 loss in PDAC patients are controversial. It was found that *SMAD4* gene inactivation, by intragenic mutation or homozygous deletion, leads to poor prognosis for patients with surgically resected pancreatic adenocarcinoma [68]. SMAD4 loss was associated with shorter OS (18.3 months versus 30.1 months for preserved SMAD4; *p* < 0.001) and disease-free survival (DFS) (6.0 months versus 13.5 months for preserved SMAD4; *p* < 0.001) in patients with resectable pancreatic cancer [69]. In advanced pancreatic cancers treated by gemcitabine, SMAD4 loss had no impact on OS (HR 1.008; *p* = 0.656) but was associated with a prolonged PFS (HR 1.565, *p* = 0.038) [70]. Recently, an analysis of publicly available datasets from The Cancer Genome Atlas (TGCA) indicated that PDAC patients with a deleted *SMAD4* gene had poorer DFS; however, *SMAD4* alterations did not predict the OS [71]. In another study, SMAD4 expression status was neither associated with early death (OR 0.5; *p* = 0.15) nor associated with recurrence pattern (OR 0.9; *p* = 0.9) in patients with resected PDAC [72]. Further investigations are needed to understand the prognostic value of SMAD4.

## 4. Discussion

### SMAD4 as a Potential Target in Patients with PDAC and Impact on Other Therapies

SMAD4 plays a central role in TGFβ signaling and seems to regulate the interruption of the cell cycle; it also improves apoptosis, decreases invasion and metastasis, and indirectly affects the prognosis of patients with PDAC. SMAD4 is, thus, designated as a potential target in the management of patients with PDAC.

A study showed that SMAD4-deficient cell lines are modestly less sensitive to gemcitabine and 2.0-fold and 4.5-fold more sensitive to cisplatin and irinotecan, respectively [73]. Controversially, a recent study found that SMAD4-deleted PDAC cells are more sensitive to gemcitabine, while *SMAD4* mutated cells had similar sensitivity to gemcitabine as *SMAD4* wild-type cells, suggesting that the SMAD4 copy number could be used as a therapeutic biomarker for PDAC treatment with gemcitabine [71]. This study also found that SMAD4-deleted PDAC cells are sensitive to other agents modulating the cell cycle including cytarabine, clofarabine, darinaparsin, and olaparib, due to the upregulation of cell cycle-related genes such as CDK1 [71]. Synthetic lethality-based screening methodologies identified two compounds (UA62001 and UA62784) to selectively target SMAD4-negative cells. In SMAD4-negative pancreatic cancer cells treated with the compound UA62001, the cell cycle was interrupted in synthesis (S) and Gap 2 (G2) – mitotic (M) phases [74], whereas UA62784 activated CDK1 and induced mitotic cell-cycle arrest and apoptosis [75]. These results confirmed results from Hsieh *et al.* and supported that SMAD4 determination status at initial diagnosis may be valuable to stratify patients for personalized chemotherapy (Figure 3). 

SMAD4 deficiency can enhance the glycolytic capacity of cancer cells with upregulation of glucose transporter expression contributing to aerobic glycolysis, which is the main metabolic pathway in PDAC [76]. PDAC harboring mutant SMAD4 exhibited a high metabolic tumor burden [66]. A subsequent study identified the glycolytic enzyme phosphoglycerate kinase 1 (PGK1) as a target for TGFβ/SMAD4. In PDAC, SMAD4 loss mediates PGK1 upregulation, which enhances glycolysis and contributes to aggressive tumor behavior. This study identified PGK1 as a decisive oncogene in patient without SMAD4 and, therefore, a potential target in the development of therapeutics for SMAD4-deficient PDAC (Figure 3) [77]. 

Different studies used a microarray-based strategy and identified miR-301a-3p, miR-483-3p, and miR-421 to target SMAD4 in PDAC. Overexpression of these microRNAs (miRNAs) promoted cell invasion, migration, and colony formation [78,79,80]. miRNA inhibitors that prevent the binding of miRNA to their mRNA targets, may have a promising effect on PDAC progression (Figure 3). Further researches need to be conducted to evaluate the therapeutic and diagnostic potential of these SMAD4 miRNA regulators.

SMAD4 mutations can interfere with its regulation associated with the Wnt/GSK3 and Erk pathways. Mutations can enhance phosphorylation by GSK3 and lead to a loss of function by protein degradation. A study suggested a therapeutic approach with GSK3 inhibitor in order to stabilize SMAD4 and restore TGFβ signaling [57]. The authors demonstrated that GSK3 inhibition reactivated TGFβ signaling in SMAD4 mutated cells in colon cancer. The potential therapeutic application of GSK3 inhibitors may be promising and requires further study in PDAC models. 

## 5. Conclusions

PDAC is projected to become the second leading cause of cancer-related death in the next decade. This alarming situation prompts the need to understand molecular mechanisms of PDAC to develop new therapies. The TGFβ signaling pathway is a powerful tumor suppressor in early stages, promoting cell-cycle arrest, differentiation, and apoptosis through SMAD-dependent signaling. In late stages, TGFβ switches its role to become a tumor promoter, favoring desmoplasia, angiogenesis, immune evasion, and EMT through MAPK, PI3K, or small GTPase pathways. SMAD4 is a key partner to TGFβ canonical signaling. In PDAC, the *SMAD4* gene is inactivated through homozygous deletion or mutation in 60% of cases. Inactivation of *SMAD4* leads to a modification in TGFβ responses, promoting non-canonical TGFβ signaling which is linked to pro-tumorigenic responses. It is still controversial whether SMAD4 loss has a real influence on the prognosis of patients with PDAC, and further investigations are needed. SMAD4 is a potential direct or indirect therapeutic target for the management of patients with PDAC, and it may also have a potential role as a therapy stratification biomarker. Patients with PDAC may benefit from SMAD4-based therapeutic strategies, but the SMAD-independent TGFβ signaling pathway may bypass strategies, leading to an innate resistance mechanism.

## Figures and Tables

**Figure 1 ijms-21-03534-f001:**
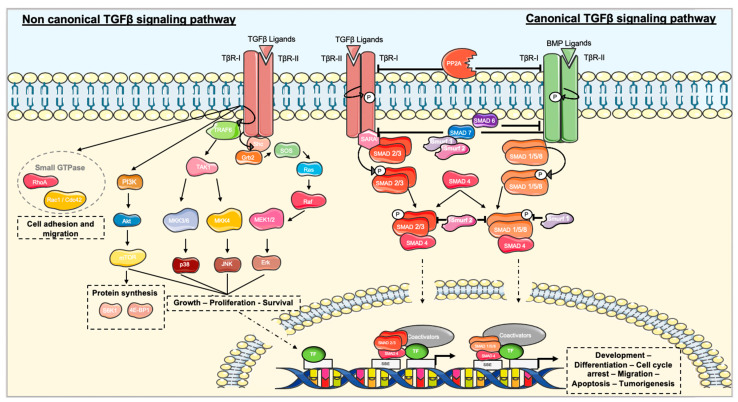
TGFβ signaling is activated through BMP or TGFβ ligands. Ligands bind the TGFβ receptor II (TβR-II) which recruits and phosphorylates TGFβ receptor I (TβR-I). In the canonical TGFβ pathway, TβR-I phosphorylates regulated SMAD (R-SMAD), SMAD2 and 3, or SMAD1, 5, and 8. SMAD4 forms a heterotrimeric complex with R-SMAD. Once translocated to the nucleus, SMAD proteins interact with transcription factors (TF) and a co-activator/co-repressor to regulate gene transcription. Signaling is stopped through inhibitory SMADs (I-SMAD), SMAD6 and 7, with the help of SMAD ubiquitination regulatory factors (Smurfs) 1 and 2. Phosphatase PP2A dephosphorylates receptors to stop the signal. Non-canonical TGFβ signaling regroups MAPKs including Erk, p38, JNK, PI3k/Akt, and the small GTPase pathway.

**Figure 2 ijms-21-03534-f002:**
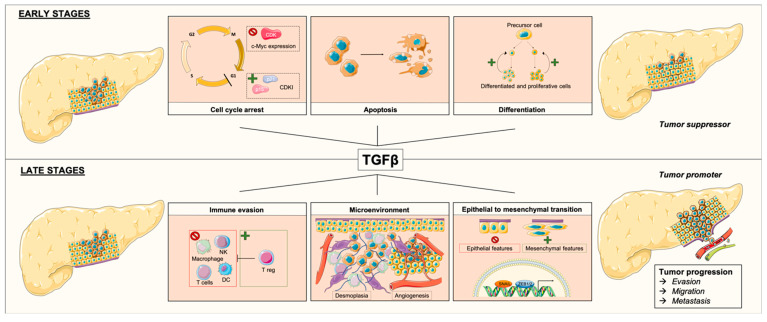
Effects of TGFβ depending of pancreatic ductal adenocarcinoma stages. At early stages, TGFβ acts as a tumor suppressor, by promoting cell-cycle arrest through repression of c-Myc expression and cyclin-dependent kinase (CDK), and through enhancement of CDK inhibitor (CDKI) expression, thereby enhancing apoptosis and stimulating differentiation of precursor cells into a less proliferative stage. At late stages, TGFβ acts as a tumor promoter through immune evasion by stimulation of regulatory T cells and by inhibition of natural killers (NKs), macrophages, T cells, and dendritic cells (DCs). TFGβ acts also on the microenvironment by promoting desmoplasia and angiogenesis, and it triggers epithelial-to-mesenchymal transition (EMT) through induction of expression of transcription factors Snail and Zeb1/2.

**Figure 3 ijms-21-03534-f003:**
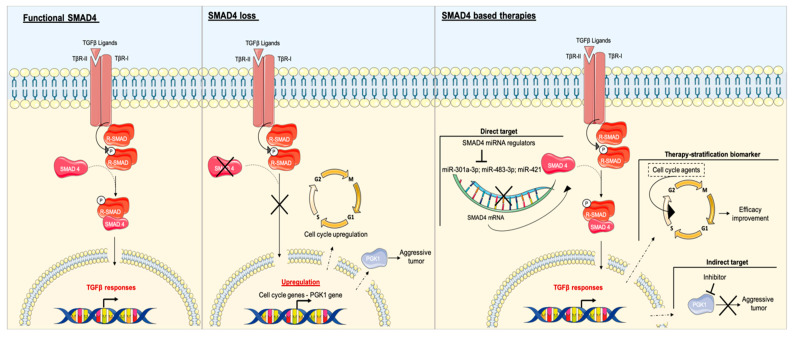
Effects of SMAD4 loss and potential therapeutic strategies in SMAD4 signaling. In normal cells, SMAD4 acts as an intracellular mediator of TGFβ. In SMAD4-null cells, loss of SMAD4 leads to upregulation of the cell cycle and phosphoglycerate kinase 1 (PGK1), leading to tumor promotion. Therapies based on SMAD4 can be direct by targeting SMAD4 miRNA regulators to modulate SMAD4 or indirect by targeting a downstream component. SMAD4 loss leads to the enhanced efficacy of agents modulating the cell cycle, whereby SMAD4 as a biomarker could be useful to stratify therapies.

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
