# Peer review of "SMAD4 and the TGFβ Pathway in Patients with Pancreatic Ductal Adenocarcinoma"

_ijms, 2020, doi:10.3390/ijms21103534_

Round 1

Reviewer 1 Report

The review is an updated window on the role of SMAD4 in the TGFβ pathway in patients with Pancreatic Ductal Adenocarcinoma (PDAC).

As written in the author's instructions, a structured review should use the same structure as research articles and ensure it conforms to the PRISMA guidelines.

Author Response

Thank you for your time and recommendation.

The article has been amended accordingly: we added a method section to explain the strategy and key words we chose to write the review. We also added results and discussion sections according to PRISMA guidelines.

Reviewer 2 Report

In this review, the authors describe the complex role of SMAD4 in the TGFβ pathway in patients with pancreatic ductal adenocarcinoma. They conclude that TGFβ can play a dual role in malignancy depending on the context, sometimes as an inhibitor and sometimes as an inducer of tumor progression. SMAD4 is a key partner to TGFβ canonical signaling. SMAD4 is a potential direct or indirect therapeutic target for the management of patients with PDAC and may also have an interest as a therapy-stratification biomarker. To my impression, this review is presented in a well-organized and logical manner. All the experimental results obtained from previous work show reasonable consistency. In addition, these studies provide insightful knowledge of SMAD4 and TGFβ and will contribute to further studies. I would therefore strongly recommend this review for publication in International Journal of Molecular Sciences.

Author Response

We thank the reviewer for this super positive feedback. Thank you for your time.

Reviewer 3 Report

In this review article the authors discuss in details the dual roles of TGF beta signaling in pancreatic cancer, with an emphasis on SMAD4. Overall, the article is nicely organized and most of the important studies have been discussed. Here are some minor issues:

Figure 2: Some legends are too small to read.

There are some typos and gramma errors such as the followings:

Page 2: In “The canonical TGF signaling pathway” paragraph:  “…bind directly to…” instead of “…bind directly…”. “R-SMADs undergo” instead of “R-SMAD undergo”.

Figure 1: “TGF signaling is activated through…” instead of “TGF signaling is expressed through…”

Page 4: the last line: “not defective” instead of “no defective”.

Page 5, Line 8: “TGF initially prevents premalignant cells but in advanced stages malignant”: this sentence is unclear. The last paragraph: “invasive behaviors” instead of “invasiveness behavior”.

Page 6, in the “SMAD4 and PDAC” paragraph: “in almost” instead of “in most of”.

Figure 3: “SMAD4 from effects to loss and therapies based”: this sentence is unclear.

Author Response

We thank the reviewer for these comments.

The manuscript has been amended accordingly.